# The Effectiveness of Vitamin E Treatment in Alzheimer’s Disease

**DOI:** 10.3390/ijms20040879

**Published:** 2019-02-18

**Authors:** Ana Lloret, Daniel Esteve, Paloma Monllor, Ana Cervera-Ferri, Angeles Lloret

**Affiliations:** 1Department of Physiology, Faculty of Medicine, University of Valencia, Health Research Institute INCLIVA, Avda. Blasco Ibanez, 17, 46010 Valencia, Spain; daniel.esteve@ext.uv.es (D.E.); paloma.monllor@uv.es (P.M.); 2Department of Human Anatomy and Embriology, Faculty of Medicine, University of Valencia, Avda. Blasco Ibanez, 17, 46010 Valencia, Spain; ana.cervera-ferri@uv.es; 3Department of Clinic Neurophysiology, University Clinic Hospital of Valencia, Avda. Blasco Ibanez, 19, 46010 Valencia, Spain; malloretalc@gmail.com

**Keywords:** Alzheimer’s disease, antioxidants, brain health, respondents to vitamin E, non-respondents

## Abstract

Vitamin E was proposed as treatment for Alzheimer’s disease many years ago. However, the effectiveness of the drug is not clear. Vitamin E is an antioxidant and neuroprotector and it has anti-inflammatory and hypocholesterolemic properties, driving to its importance for brain health. Moreover, the levels of vitamin E in Alzheimer’s disease patients are lower than in non-demented controls. Thus, vitamin E could be a good candidate to have beneficial effects against Alzheimer’s. However, evidence is consistent with a limited effectiveness of vitamin E in slowing progression of dementia; the information is mixed and inconclusive. The question is why does vitamin E fail to treat Alzheimer’s disease? In this paper we review the studies with and without positive results in Alzheimer’s disease and we discuss the reasons why vitamin E as treatment sometimes has positive results on cognition but at others, it does not.

## 1. Alzheimer’s Disease and the Hypothesis of Its Onset

Alzheimer’s disease (AD) is a neurodegenerative disorder characterized by a long evolution whose clinical symptoms appear late in life. However, in the last years the paradigm of AD has changed. In the past, researchers thought AD was an age-related disorder which begins during the aging process. Today we know that the onset of the disease occurs between 15 (for the genetic) and 20–30 years (for the sporadic) before any clinical symptom appears [1]. There is no a preventive or curative therapy for the disease and the lack of knowledge of when the disease begins greatly complicates the work of the physicians. Another added handicap is that neither do we know why the disease begins. 

In this sense, there are several hypotheses trying to explain the beginning of AD. These hypotheses may not be exclusive, and they may well overlap and take place at the same time. We can divide the hypotheses into three groups: The hypotheses based on protein deposits. This group includes the beta-amyloid (Aβ) cascade hypothesis; and the tau hypothesis. 

The deposits mainly formed by Aβ peptide are known as senile plaques [2]. Aβ comes from the proteolysis of a membrane protein called amyloid precursor protein (APP). In favor of the Aβ cascade theory we can say that mutations in genes involved in the genesis of Aβ cause AD [3,4]; mutations in the gene encoding the tau protein do not cause amyloid deposition [5,6]; the ApoE4 allele leads to a reduction in the clearance of the Aβ peptide and increases the risk of AD [7]; Aβ oligomers that are isolated from AD brains involve loss of synapses, neuronal density, and memory impairment [8]; Aβ peptide can induce hyper-phosphorylation of tau [9].

The deposits essentially formed by the tau protein are known as neurofibrillary tangles. Tau is a cytoskeleton protein which is very important for its stability. Tau changes to a hyper-phosphorylated state causing a disruption of the cytoskeleton in AD pathology. Neurons with a high content of hyper-phosphorylated tau enter into apoptosis and neurodegeneration takes place [10]. In favor of this theory, we can say that the severity of this type of dementia correlates well with the growing accumulation of neurofibrillary tangles in the brain [11,12,13]; there is a high correlation between hyper-phosphorylated tau species in the cerebrospinal fluid (CSF) in patients with AD and the degree of cognitive impairment [14]; a decrease in tau filaments by drugs directed against this therapeutic target alleviates cognitive deterioration [15].

The hypothesis of reactive processes which includes neuroinflammation as the first event in AD. An elevation of proinflammatory cytokines are found in AD [16,17]. High levels of tumor necrosis factor alpha (TNF-α) and interleukin 6 (IL-6) are also detected in the serum and in brain of patients when compared to controls. Multiple inflammatory markers are found in AD animal models, such as IL-1, IL-6, the granulocyte-macrophage colony-stimulating factor (GM-CFS), IL-12, IL-26, and TNF. Histologically, characteristic amyloid plaques are surrounded by microglia and reactive astrocytes appear in the brains of patients with AD [18]. Moreover, according to studies with mice with cerebral amyloidosis, the activation of astrocytes seems to occur very early in the pathogenic process [19]. Specifically, it has been seen that the elevation of both cells and proinflammatory cytokines appears before the deposit of Aβ [20]. 

The hypotheses based on loss of function: calcium misbalance hypothesis, vascular hypothesis and oxidative stress hypothesis. In favor of the calcium theory we can say that it has been confirmed that each mutation of early-onset AD alters the calcium balance in the cell [21]. In sporadic AD, before the presence of Aβ or any other histological alteration, there is evidence suggesting neuronal hyper-excitability due to an increase in calcium levels in neurons [22].

In favor of the vascular theory we can say that there is a clear relationship between vascular risk factors and AD; early vascular damage is present in individuals with AD; vascular damage alone causes neurodegeneration; there are intravascular deposits of Aβ in early phases of AD [23].

Lastly, one of the most important the accumulated experimental evidence is the oxidative stress theory. 

Despite of the different theories about the onset of AD there is a common thread in all of them that occurs in the ageing cells causing, among other effects, a poor functioning of energy metabolism. This thread is oxidative stress and its role in this disease is central. We will discuss it in a broad manner in the following lines of this review and will interrelate its role with the role of vitamin E which is, mostly, tackling the effects of oxidative stress. 

## 2. Oxidative Stress Theory and AD

Oxidative stress is the imbalance between high oxidant species formed and insufficient antioxidant defenses [24]. This causes an altered homeostatic balance resulting from oxidant insult [25]. The reduction of O_2_ to H_2_O in mitochondria. The reduction of O_2_ to H_2_O in mitochondria produces reactive oxygen species (also known as ROS) which are highly reactive and interact with several biological molecules nearby. On the other hand, the brain is the organ with the highest O_2_ consumption in our body. Despite composing the 2–3% of the total body weight, it uses about 20–25% of the basal metabolism. As a consequence, it is the highest ROS-generator organ. The brain metabolism is early reduced in AD and this could be associated with a decreasing metabolic regulation and increased ROS generation [26].

In AD, oxidation of all macromolecules is found very early in the brain of patients. Lipids, proteins, nucleic acids, and polysaccharides are all oxidized in AD. 

Advanced glycation end products (AGE) are formed by glycation, a reaction between reduced sugars and protein side chains. Glycation products are very stable molecules and trend to accumulate inside neurons, in senile plaques and in neurofibrillary tangles [27]. 

Lipid peroxidation consists in the hydroxy radical attack of unsaturated lipids to generate highly reactive secondary products such as reactive carbonyls and reactive aldehydes, which are able to inactivate enzyme active sites overriding their physiological role. Furthermore, oxidized membranes have altered mobility. Aldehyde adducts of protein are common on senile plaques and neurofibrillary tangles and are most prominent in cell bodies of vulnerable neurons [28].

When proteins are oxidized, the peptide bond could be compromise and it can be cleaved. Moreover, reactive carbonyls are frequently generated and also protein nitration, a related phenomenon. All these protein modifications happen prominently in neuronal cell bodies [29].

Lastly, nucleic acids could also be affected by oxidation. ROS can alter both purinic and pyrimidinic bases with mutagenic and even deleterious results. Vulnerable neuronal cell bodies accumulate ostensibly oxidized nucleic acids [30].

Consequently, it is not surprising that antioxidant therapy has been proposed countless times as a treatment for Alzheimer’s disease. 

## 3. Why Vitamin E as Treatment for AD?

### 3.1. Vitamin E Is an Antioxidant and Neuroprotector

Vitamin E is a term that includes a group of eight compounds and belongs to the fat-soluble vitamins. In this group we can find α-, β-, γ-, and δ-tocopherols and tocotrienols, their main characteristic is their antioxidant potential. However, vitamin E also has neuroprotective, anti-inflammatory, and hypocholesterolemic properties [31,32,33] driving to its importance for brain health.

The antioxidant role of vitamin E is based on the presence of a hydroxyl group in its phenolic group on the chromanol ring that can donate a hydrogen atom and, in this way, neutralize a great variety of free radicals including reactive oxygen species (ROS) [34]. When this reaction takes place, a non-radical product and a vitamin E radical are obtained. Then the vitamin E radical can react with another free radical lipid or can be regenerated back to their native form by vitamin C [35,36,37,38]. By this process vitamin E neutralizes the peroxyl radicals and blocks lipid peroxidation [39], especially the polyunsaturated fatty acids peroxidation, which is essential for the protection of cellular membranes. 

Vitamin E has more antioxidant potential against peroxyl radicals than other antioxidants like glutathione or β-carotene [40]. This antioxidant activity has been proved in several studies both in vitro and in vivo. One of the most important pieces of evidence of the antioxidant capacity of vitamin E was reported on 1997 by Ham and Liebler who gave supplementary vitamin E diet to rats. They evaluated the antioxidant properties of vitamin E inducing lipid peroxidation in liver cells by t-Bu-OOH. They showed a reduction in the metabolic changes in vitamin E-treated rats [41].

Moreover, vitamin E is considered one of the most important antioxidants in the brain and especially the α-tocopherol form. This is due to the high levels found in the brain of its transporter α-TTP (α-tocopherol transfer protein) whose functions include the regulation and distribution of levels of vitamin E in different tissues [42,43]. Its critical role in brain function is underscored by the fact that human carriers of a mutation in the α-TTP gene develop progressive spinocerebellar ataxia, areflexia, loss of proprioception, and extremely low vitamin E levels [44,45]. In the encephalon, α-TTP expression is highest in the cerebellum, specifically in astrocytes, which provide vitamin E to the neighboring neurons [46]. Importantly, α-TTP expression is increased in brains of patients with neurodegenerative diseases [42,47]. Therefore, we can conclude that vitamin E plays an important role in the neuroprotection through its antioxidant activity. 

As mentioned previously, in AD there is an evident production of ROS that promotes the oxidation of all macromolecules. It may not be surprising the fact that Aβ is by itself an important inductor of oxidative stress inside cells since it involves generation of ROS by mitochondria and endoplasmic reticulum. Aβ also causes a disruption of cellular calcium homeostasis, crucial for the transmission of action potential. When Aβ aggregates and deposits near the cell membrane, it can promote lipid peroxidation to produce pro-oxidant species, including malondialdehyde (MDA) or 4-hydroxynonenal (4HNE). The last one, is an aldehyde able to covalently modify lipids and proteins on its vicinity and one of these proteins is tau [48,49].

Oxidative modification of tau can promote its aggregation in vitro [50] and may induce tau hyper-phosphorylation and the formation of neurofibrillary tangles [51]. Additionally, oxidative stress also favors tau hyper-phosphorylation by promoting glycogen synthase kinase-3 beta (GSK3β) activity. GSK3β is a ubiquitously expressed kinase and it phosphorylates tau in most serine and threonine residues in paired helical filaments [52]. Finally, 4HNE directly activates a mitogen-activated protein (MAP) kinase, p38 [53], which leads to tau hyper-phosphorylation. Truthfully; a correlation was found in transgenic mice exhibiting hyper-phosphorylated tau between activated p38 and the level of aggregated tau [54].

Lastly, vitamin E is considered one of the most important antioxidants in the brain and especially the α-tocopherol form. This is due to the high levels found in the brain of its transporter α-TTP (α-tocopherol transfer protein) whose functions include the regulation and distribution of levels of vitamin E in different tissues [42,43]. Its critical role in brain function is underscored by the fact that human carriers of a mutation in the α-TTP gene develop progressive spinocerebellar ataxia, areflexia, loss of proprioception, and extremely low vitamin E levels [44,45]. In the encephalon, α-TTP expression is highest in the cerebellum, specifically in astrocytes, which provide vitamin E to the neighboring neurons [46]. Importantly, α-TTP expression is increased in brains of patients with neurodegenerative diseases [42,47]. Therefore, we can conclude that vitamin E plays an important role in the neuroprotection through its antioxidant activity. 

### 3.2. Vitamin E as an Anti-Inflammatory and Cell Signaling

Beyond its role as an antioxidant, vitamin E can enhance the immune response in elderly people. During aging and in some neurodegenerative diseases such as AD, a dysregulation of the immune system which triggers many inflammatory responses are produced. There are many studies that prove the anti-inflammatory property of vitamin E when elderly people take it as a diet supplement. 

These studies show in vitro T cells proliferation, IL-2 production and E2 prostaglandin inhibition among other beneficial effects [32]. With 233 mg of vitamin E every day for 28 days, Lee and co-workers found an increase in IL-2 receptor levels in T-lymphocyte population [55]. With a similar dose during three months, De La Fuente et al. showed a beneficial effect of α-tocopherol in elderly people, specifically in the adherence capacity of lymphocytes, in increased IL-2 production, in NK activity and in lymphocytes proliferation [56]. 

However, this beneficial effect disappears with a concomitant disease, like allergic rhinitis [57] or causes a hypersensitivity response [58] or even harmful effects in adult smokers [59,60].

The action mechanism of vitamin E on the immune system involves the inhibition of prostaglandins E2 and D2 without inhibition of cyclo-oxygenase (COXs) and 5-lipoxygenase (5-LOX) activities [61,62,63]. On the one hand, vitamin E metabolites from the ω- and β-oxidation of its hydrophobic side chain inhibit COX activity [62]. On the other hand, vitamin E impedes membrane changes which trigger in the blockage of the calcium influx, through the inhibition of ionophores that produce the inhibition of 5-LOX activity [63]. Both inhibitions activate a signaling pathway that finally induces the inhibition of the prostaglandins. Then IL-2 production is generated and, therefore, an immune response (Figure 1).

It has been shown that vitamin E is an inhibitor of the protein kinase C (PKC) and this inhibition is independent of its antioxidant activity [64]. The mechanism is doubled: vitamin E can activate the phosphatase (PP)2A that inhibits the active form [65,66], and vitamin E can modulate the diacylglycerol kinase activity [65,66]. The result is different depending on the vitamin E isoform, α-tocopherol inhibits PKC in the vascular smooth muscle and cell growth is arrested whereas β-tocopherol prevents this effect [64,67,68]. 

By contrast, the inhibition of PKC by vitamin E could have non-beneficial effects in some aspects of AD pathology. For example, PKC plays an important role in immune response because T-cell subpopulation reactive to Aβ1-42 expresses different isoforms of PKC at different clinical stages of AD [69]. Moreover, PKC regulates α-secretase activity which is essential in the non-amyloidogenic APP processing. Thus, the amyloidogenic via is reduced and, therefore, the formation of amyloid beta toxic peptide [70]. 

Vitamin E can also regulate genes at a transcriptional level in a PKC-independent way, some of these genes are CD36 [71], SR class A [72] and intracellular adhesion molecule-1 [73]. 

To summarize, vitamin E not only acts as an antioxidant but also has cell signaling properties, both in a PKC-dependent or PK-independent way. Thus, the study of vitamin E treatment in AD should not only focus on its antioxidant properties, but also on anti-inflammatory and neuroprotection effects (Figure 2). 

### 3.3. Levels of Vitamin E in AD Are Low

Nearly 30 years ago, it was published that vitamin E levels were decreased in 55 patients with AD compared to non-demented controls [74]. Since then, many works have corroborated these results. Sometimes with low number of patients (see Table 1), but always with significant results. A meta-analysis performed in 2014 reviewed 80 studies on micronutrients and AD. The authors concluded that vitamin E, among others, showed lower plasma levels in AD patients. Moreover, the authors did not find a relation between the levels of vitamin E and the malnourishment status of the patients and they suggest that micronutrient status may be compromise before malnutrition [75]. A recent meta-analysis of 116 selected publications has confirmed that vitamin E levels in AD patients are also significantly lower in CSF and in brain [76]. A more recent meta-analysis using 17 studies and including an overall of 904 AD patients and 1153 controls showed that AD patients had lower concentration of serum Vitamin E compared with healthy older controls [77].

However, although they are less numerous, we can also find other studies reported no difference in vitamin E levels (Table 2). A very recent Mendelian randomized study investigated the relationship between circulating vitamin E and AD. For this purpose, the authors used the data from a large-scale vitamin E GWAS (genome-wide association study) with a total number of 7781 individuals of European descent and also a GWAS that includes 17,007 AD cases and 37,154 controls. This study showed no significant correlation between vitamin E levels and AD [88].

### 3.4. Vitamin E and Prevention of Cognitive Decline

Regarding epidemiological studies an association between supplementation with vitamin E and decreased risk of developing AD was also revealed. In 1998, a prospective study was published with 633 persons where none of the 27 vitamin E supplement users had AD after a period of 4.3 years of follow-up [89]. In 2002, Engerhalt et al. corroborated this result in a different cohort from The Netherlands, with a six-year follow up [90]. Another prospective study was published the same year, conducted from 1993 to 2000, of 815 residents aged 65 years and older, free of AD at baseline, and were followed up for a mean of 3.9 years. This study suggested that food containing vitamin E, but not other antioxidants, may be associated with a reduced risk of developing AD. However, this association was observed only among the individuals not carrying the APOE ɛ4 allele [91]. Another prospective study, the Cache Country Study from Utah (USA) concluded an evident lower AD risk in people supplemented with vitamin E and multivitamin complexes containing, among others, vitamin C. Interestingly, they showed no evidence of protective effect associated to the intake of these compounds alone [92]. 

Analyzed data from the Rotterdam Study (365 AD patients) also showed a modest reduction in the long-term risk of AD. Importantly, the positive results were only present in those participants with higher intake of foods rich in vitamin E. However, participants with average vitamin E intake did not have a lower risk of dementia [93]. Data from the Canadian Study of Health and Aging (1991–2002), a cohort study of dementia including 560 AD patients, suggested that the use of vitamin E supplements was associated with a reduced risk of cognitive decline [94]. 

On the other hand, three other studies did not show any association between vitamin E intake and risk of suffering AD. The first, with 2969 participants followed up biennially for 5.5 years, concluded that the use of supplemental vitamin E and C, alone or in combination, did not reduce risk of AD or overall dementia [95]. The second, with 3385 men from The Honolulu–Asia Aging Study, found that vitamin E and C supplements may improve cognitive function in late life, but no protective effect was found for Alzheimer’s dementia in particular [96]. Lastly, another study with 980 elderly subjects in the Washington Heights-Inwood Columbia Aging Project, found that neither dietary, supplemental nor total intake vitamin E was associated with a decreased risk of AD [97]. 

**Table 2 ijms-20-00879-t002:** Studies that don’t show a relation between AD and vitamin E levels.

Authors and Publication Year	Isoform	Method	Number of Patients and Diagnosis	Results
Schippling et al, 2000 [98]	α-tocopherol	HPLC	29 patients	The difference of α-tocopherol levels among AD patients and controls were no significant
von Arnim et al., 2012 [99]	α-tocopherol	HPLC	74 MCI patients	No association was found for vitamin E levels and dementia
Charlton el at., 2004 [100]	α-tocopherol	HPLC	15 AD patients	No differences in the vitamin E levels between AD patients and controls
Ryglewicz et al., 2002 [101]	α-tocopherol	HPLC	26 AD patients	Levels of vitamin E was significantly lower in patients with vascular dementia in comparison to patients with AD and controls

## 4. Is Vitamin E Effective as Treatment for AD? An Approach to Main Trials

An early study published in 1997 by Sano et al. showed the efficacy of vitamin E as treatment in AD for the first time. This double-blind, randomized multicenter clinical trial was also placebo-controlled and mainly focused on vitamin E supplementation in AD patients with moderately severe impairment [102]. Three hundred and forty-one patients were recruited and received 2000 IU/d vitamin E or placebo for two years [102]. As results, they measured time to occurrence of death, institutionalization, loss of ability to perform basic daily living activities, or severe dementia [102]. They concluded that this dose of vitamin E slows AD progression. Twenty-two years later the controversy still exists. Table 3 summarizes Clinical trials about the effectiveness of vitamin E in AD treatment.

In 2005, Petersen et al. selected 769 subjects and performed a double-blind study [103]. The subjects received 2000 IU/d of vitamin E or placebo for three years, but the authors found that vitamin E supplementation had no benefit [103]. Interestingly, a study published in 2009 (800 IU/d, six months) found two different subgroups in the vitamin E supplemented group. In one group, called “respondents” to vitamin E, oxidative stress parameters were lower after the treatment and scores on the cognitive tests were maintained. However, the authors found a second group called “non-respondents”, in which vitamin E was not effective in preventing oxidative stress and cognition decreased to levels even lower than those of patients taking placebo. They concluded that when vitamin E lowers oxidative stress cognitive status is maintained in patients. However, when vitamin E does not prevent oxidative stress, it is detrimental in terms of cognition [104].

Nonetheless, in 2014, Dysken et al. in a double-blind, placebo-controlled, parallel-group, randomized clinical trial (The TEAM-AD VA Cooperative Randomized Trial) recruited 613 patients with mild to moderate AD and with the same dose (2000 IU/d of alpha-tocopherol or placebo) found that α-tocopherol supplementation reduced functional decline in patients with mild to moderate AD [105]. Curiously, these results were not observed with memantine or with the combination of memantine and vitamin E treatment, so it could be that memantine interferes in the vitamin E effect [105]. 

In 2017 in the PREADViSE study, Kryscio et al. with a low dose of vitamin E (400 IU/d) found that vitamin E supplementation in selected 7540 asymptomatic older men did not prevent dementia [106]. They evaluated the effect of the vitamin E and selenium supplementation in the prevention of AD and concluded that neither vitamin E, nor selenium, nor even a combination of both prevented dementia [106].

However, the attempts to obtain an effective treatment for AD based on vitamin E has continued. There are still many studies that assess the effectiveness of different forms of vitamin E or the combination of vitamin E with another compound that could have a beneficial effect against the evolution of dementia. Last year, the group headed by Ikuo Tooyama evaluated the beneficial effect of the tocotrienol-rich fraction, a combination of different vitamin E analogues from palm oil, in a transgenic mice model of AD [107]. The results showed that the supplementation rescued the cognition function in the transgenic mice and reduced the Aβ deposition although the Aβ oligomers levels did not change [100]. Finally, another study with an AD animal model evaluated a combination of vitamin E with fish oil supplementation [108]. Their results showed that only a low vitamin E diet and fish oil supplement rescued the cognitive function in the transgenic mice. Higher doses of vitamin E were not beneficial suggesting that high levels of vitamin E act as a prooxidant causing oxidation [108]. 

## 5. Why Does Vitamin E Fail to Treat AD?

Since oxidative stress is an early hallmark of AD, many efforts have been made by the scientific community to try antioxidant treatment as a possible therapy. However, as we have discussed above, evidence is consistent with a limited effectiveness of vitamin E in slowing progression of dementia; the information is mixed and inconclusive. However, why does vitamin E with other antioxidants fail to improve the course of cognitive decline in AD? This highlights a paradox in oxidative stress: the efforts to increase oxidative defenses have not proved beneficial for human diseases [109] which has been called “antioxidant paradox”. Halliwell points out three possibilities: the extrapolation of results obtained in animals to humans, the lack of measurements of baseline nutritional status in human cohorts and some antioxidants can have pro-oxidant effects [110]. As we have discussed above, plasma levels of vitamin E are low in the patients, so we have evidence about the “baseline nutritional status”. However, it is true that the measurement of baseline vitamin E levels is often not check in the main trials before beginning.

On the other hand, the extrapolation of results from animals to humans is not trivial. The correct methods to measure oxidative stress are still under debate and sometimes are not specific or appropriated [111,112,113,114]. Many studies of oxidative stress in AD are old and methods could not be up-to-date. To follow the antioxidant response by measuring an oxidative stress biomarker is very recommendable. However, to choose a good oxidative stress biomarker could not be easy. The specificity of some used biomarkers of oxidative stress, as in the case of oxidized low-density lipoprotein antibodies, could be questionable. In this regard, it has been recommended the analysis of oxidative stress in specific proteins involved in the disease. These markers could better represent a specific pathological pathway and a way for therapeutic monitoring and a prediction of results [115]. An example of the importance of monitoring oxidation in patients is the ineffectiveness of vitamin E supplements in modifying the oxidative balance, as in the non-respondents group found by Lloret et al. [104]. In this work was found that only patients in whom oxidative stress had decreased showed a cognitive improvement, but non-respondents worsened in their results of cognitive tests.

Another important point is bioavailability of vitamin E, which is complex and influenced by some important factors, such as the intake of competing nutrients, intestinal differences in absorption, age, gender, smoking, obesity and genetic polymorphisms. Sterols from plants, eicosapentaenoic and retinoic acids, and even dietary fiber, are described as competing nutrients that decrease the absorption of vitamin E [116,117,118,119]. Moreover, the rate of absorption into the bloodstream differs between 20% and 80% in humans, mainly due to the differences in food contents and the form of vitamin E. The transport of vitamin E in blood circulation is done by lipoproteins [120]. In particular, the entrance to the central nervous system of α-tocopherol in vivo is provided by high-density lipoprotein (HDL) particles and HDL levels vary greatly among people [121]. In cells, vitamin E is bound to α-TTP, a transport protein highly expressed in AD brains and especially under oxidative stress conditions [47].

Age and gender also influence vitamin E bioavailability. After 60 years the levels of vitamin E increase in plasma but decrease after 80 years [122]. Women seem to have higher maximum plasma concentrations of α-tocopherol than men maybe due to differences in HDL levels between genders. [123].

On the other hand, smokers have lower levels of a-tocopherol in serum than non-smokers, although the variances could be attributed to the differences in dietary patterns [124,125,126]. And obesity has an inverse relationship with serum α-tocopherol levels [127]. In particular, the waist-to-hip ratio and waist circumference were associated with α-tocopherol serum concentrations independently of gender. [128,129,130].

Lastly, genetic polymorphisms in genes encoding for proteins involved in vitamin E metabolism could also explain interindividual differences in bioavailability [131,132,133,134]. More than 50 different single-nucleotide polymorphisms (SNPs) has been related to vitamin E homeostasis. 

Another point is the efficacy of vitamin E to pass through the blood-brain barrier and the raise of levels in brain. Therefore, demonstrating the efficacy of vitamin E in reducing brain oxidative damage is very important [135].

Many other studies can be confusing due to the form of vitamin E supplemented or due to the combination with other antioxidants, nutrients and pharmaceuticals used [136].

The Mediterranean diet along with moderate physical activity is nowadays believed to be a first line of defense against the development and progression of AD. However, in most cases the studies defending this idea were observational. As a consequence, results from large, multicenter randomized clinical trials clarifying the actual link between moderate exercise, physical activity and healthy Mediterranean diet on cognition in the elderly are awaited. It is true that an antioxidant-enriched diet may be better than pure antioxidants intake in a pharmaceutical form, because their bioavailability could be different [137].

Another crucial fact is the complexity of the brain. Oxidative stress leads to neuronal damage and even apoptosis cell death and neurodegeneration, so neuronal networks change, and compensatory response is induced. Reductive stress rather than oxidative stress was found in healthy young person ApoE 4/4 [138]. This scenario complicates enormously the successful efficacy of antioxidant therapy. Remodel or regenerate the lost neuronal network properly is highly difficult and ever more so if we consider that it is an aging brain or a compensating brain. 

## 6. Conclusions

At the present time, clinical studies have revealed unreliable findings on the effect of vitamin E on AD-developing risk. Thus, it remains unclear whether vitamin E levels are genetically associated with AD risk or if the supplementation with this compound could be beneficial in delaying the progression of dementia. In this review we can find that the number of studies showing a decrease in the levels of plasma vitamin E in AD patients is higher than the studies with contrary results. However, vitamin E as treatment sometimes has positive results on cognition but at others, it does not. The loss of neuronal networks and its replacement, the very different nutritional status of the patients at baseline, the range of time of brain compensation in each person, the antioxidant effect of vitamin E in each person among others could be the reason for the failure in the treatment. 

## Figures and Tables

**Figure 1 ijms-20-00879-f001:**
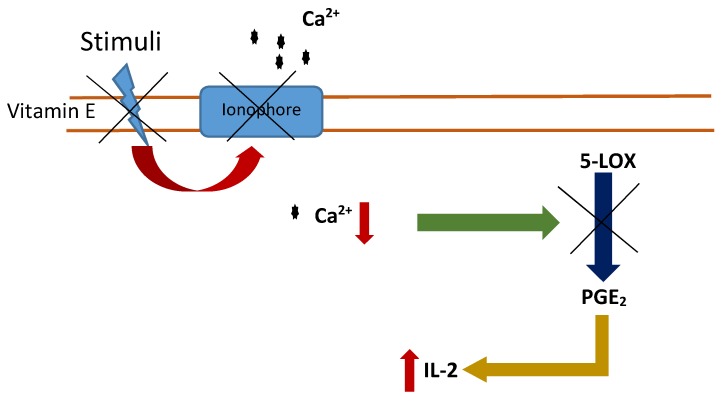
Inhibition of 5-LOX activity by vitamin E. Vitamin E blocks calcium ionophores inducing a reduction in the intracellular calcium levels, triggering the inhibition of 5-LOX activity and inducing the inhibition of the prostaglandins (PGE_2_). Interleukin 2 (IL-2) levels increase and consequently an immune response.

**Figure 2 ijms-20-00879-f002:**
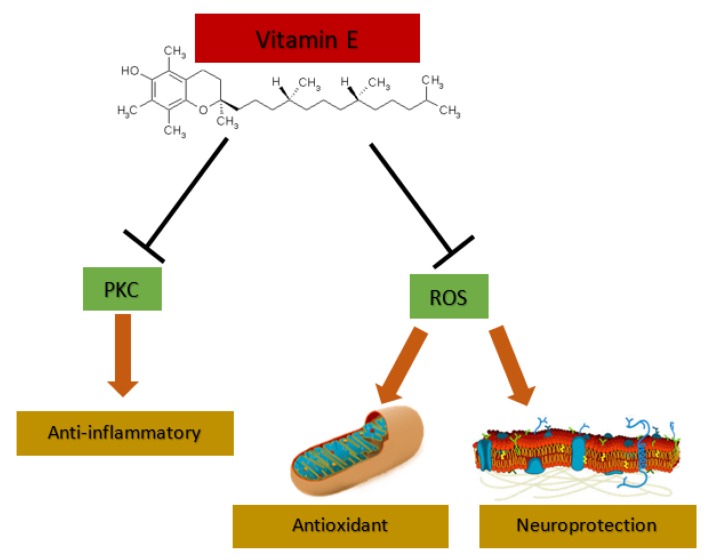
Vitamin E effects. Vitamin E can act as anti-inflammatory through protein kinase C (PKC) inhibition and has antioxidant and neuroprotection properties through the attack to reactive oxygen species (ROS).

**Table 1 ijms-20-00879-t001:** Studies about the relation of Alzheimer’s disease and reduction of the vitamin E levels.

Authors and Publication Year	Isoform	Method	Number of Patients and Diagnosis	Results
Zaman et al., 1992 [78]		High-performance liquid chromatography (HPLC)	10 AD patients	Lower levels of plasma vitamin E
Jimenez-Jimenez et al., 1997 [79]	α-tocopherol	HPLC	44 AD patients	Decreased levels of vitamin E both in serum and in CSF
Sinclair et al., 1998 [80]	α-tocopherol	HPLC	25 AD patients	Lower plasma levels of vitamin E
Foy et al., 1999 [81]	α-tocopherol	HPLC	79 patients with AD	Lower plasma levels
Bourdel-Marchasson et al., 2001 [82]	α-tocopherol	HPLC	20 patients	Lower plasma levels of α-tocopherol
Polidori et al., 2002 [83]	α-tocopherol	HPLC	35 patients	Plasma concentration of vitamin E lower and malondialdehyde higher
Rinaldi et al., 2003 [84]	α-tocopherol	HPLC	25 patients with mild cognitive impairment (MCI) and 63 AD patients	Vitamin E decrease
Mecocci et al., 2002 [85]	α-tocopherol	HPLC	40 patients	Vitamin E decrease in plasma
Giavarotti et al., 2013 [86]	α-tocopherol	Precolumn and reverse phase C18 column and UV-VIS detector with deuterium lamp, a mobile phase with 80% acetonitrile, 3% methanol and 15% dioxan	23 patients with AD	Lower plasmatic levels of α-tocopherol
Mullan et al., 2017 [87]	α- and γ-tocopherol	HPLC	251patients with AD	Lower levels of α-tocopherol but γ-tocopherol higher in serum of AD patients

**Table 3 ijms-20-00879-t003:** Clinical trials about the effectiveness of vitamin E in the AD treatment.

Authors and Publication Year	Number of Patients and Diagnosis	Isoform	Doses	Time	Method	Results
Sano et al., 1997 [102]	341 AD	α-tocopherol	2000 IU/d for 2 years	Two years	Alzheimer’s Disease Assessment Scale (ADCS); Mini–Mental State Examination (MMSE); Blessed Dementia Scale; Dependence Scale	Decline progression of AD
Petersen et al., 2005 [103]	769 AD	not specified	2000 IU/d for 3 years	Three years	A battery of fifteen cognitive tests (MMSE, The Clinical Dementia Rating (CDR), ADCS…)	No benefit
Lloret et al., 2009 [104]	33 AD	α-tocopherol	800 IU/d for 6 months	Six months	MMSE; Blessed-Dementia Scale; Clock Drawing Test	Cognitive status was maintained in some cases but in others it was detrimental in terms of cognition
Dysken et al., 2014 [105]	613 mild to moderate AD	α-tocopherol	2000 IU/d	6 months–4 years	Alzheimer’s Disease Cooperative Study/Activities of Daily Living (ADCS-ADL) Inventory; MMSE	Reduced functional decline
Kryscio et al., 2017 [106]	7540 asymptomatic older men		400 IU/d	6 years	Memory Impairment Screen (MIS); Consortium to Establish a Registry for AD (CERAD) battery	No prevention of dementia

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
