# Peer review of "The Effectiveness of Vitamin E Treatment in Alzheimer’s Disease"

_ijms, 2019, doi:10.3390/ijms20040879_

Round 1

Reviewer 1 Report

The manuscript: "The effectiveness of vitamin E treatment in Alzheimer’s disease" by Ana Lloret and colleagues reviews different studies with and without positive results of Vitamin E treatment in Alzheimer’s disease. The manuscript is well-written and is informative with elaborate data giving a good overview of different studies.  Since the manuscript is a review of different studies, it does not conclusively explain why Vitamin E has shown positive effect in Alzheimer’s patients in some studies and no positive effect in other studies. This point is still debatable and a conclusive explanation is of yet not available. Hence, I would not like to criticize the manuscript and would like to suggest accepting the manuscript in present form.

Author Response

We thank the reviewer for his/her kind comments.

Reviewer 2 Report

The authors wrote the review article about the effectiveness of vitaminE for the treatment of AD. Vitamin E is an antioxidant and neuroprotector

and it has anti-inflammatory and hypocholesterolemic properties. This review article is well written, and interesting.  

The reviewer is qurious about the possible interaction of vitamin E and phosphorylation level of tau protein. If there are previous reports about vitamin E can reduce the phosphorylated tau protein, please describe in the discussion section.

Author Response

Reviewer 2: The authors wrote the review article about the effectiveness of vitaminE for the treatment of AD. Vitamin E is an antioxidant and neuroprotector and it has anti-inflammatory and hypocholesterolemic properties. This review article is well written, and interesting.  The reviewer is qurious about the possible interaction of vitamin E and phosphorylation level of tau protein. If there are previous reports about vitamin E can reduce the phosphorylated tau protein, please describe in the discussion section.

We thank the reviewer for this helpful comment. We have added the following paragraph to the manuscript:

“As mentioned previously, in AD there is an evident production of ROS that promotes the oxidation of all macromolecules. It may not be surprising the fact that Aβ is by itself an important inductor of oxidative stress inside cells since it involves generation of ROS by mitochondria and endoplasmic reticulum. Ab also causes a disruption of cellular calcium homeostasis, crucial for the transmission of action potential. When Aβ aggregates and deposits near the cell membrane, it can promote lipid peroxidation to produce pro-oxidant species, including malondialdehyde (MDA) or 4-hydroxynonenal (4HNE). The last one, is an aldehyde able to covalently modify lipids and proteins on its vicinity and one of these proteins is Tau (Mattson MPO, 2004; Alavi Naini and Soussi-Yanicostas, 2015).

Oxidative modification of Tau can promote its aggregation in vitro [Gamblin et al, 2000] and may induce tau hyper-phosphorylation and the formation of neurofibrillary tangles [Perez et al, 2000]. Besides, oxidative stress also favours tau hyper-phosphorylation by promoting GSK3β activity. GSK3β is a ubiquitously expressed kinase and it phosphorylates tau in most serine and threonine residues in paired helical filaments [Hernandez el at, 2013]. Finally, 4HNE directly activates p38 MAP kinase [Kelleher et al, 2007] which leads to tau hyper-phosphorylation. Truthfully; a correlation was found in transgenic mice exhibiting hyper-phosphorylated tau between activated p38 and the level of aggregated tau [Giraldo et al, 2014].”

Reviewer 3 Report

At first glance I was very much looking forward to the promise announced in the abstract; reasons why vitamin E can have positive and adverse effects in the treatment of AD (by the way, treatment was not described, rather prevention of development). However, reading the following was disappointing. The focus was laid only on the involvement of oxidative stress (free radicals) in the development of AD and the antioxidative property of vitamin E. (Free radical scavenging by vitamin E was never proved in vivo!) Related references were mainly derived from the last millennium. Thus, novelties in this direction were scarce. Methods used about 20 years ago to find oxidative stress as inducer or mediator of diseases were not specific and are no longer up-to-date (see references below). The focus on redox events in disease development is now concentrated on the redox-regulated modification of cellular signaling cascades involved in disease development or prevention. Modification of signaling pathways might well be a mechanism for the change in the other events hypothesized to start the AD development. According to the hypotheses provided in the introduction, the reader would expect a discussion in this direction.

Nevertheless, presenting the summary of available studies would help to further dig into the data and find a possible reason for the inconsistency of vitamin E effects. For this, the form of vitamin E, the doses, the time of intervention and the analytical methods should be included in all tables.

In sum, in the present form, the manuscript cannot be recommended for publication. However, it might become more interesting if the state of the art is better presented.

References

Kalyanaraman, B., Darley-Usmar, V. Davies, K.J., Dennery, P.A., Forman, H.J., Grisham, M.B., Mann, G.E., Moore, K. Roberts II, L.J.,  Ischiropoulos, H.

Measuring reactive oxygen and nitrogen species with fluorescence probes:  challenges and limitations. Free Radic. Biol. Med. 52:1-6, 2012.

Winterbourn CC

The challenges of using fluorescent probes to detect and quantify specific oxygen species in living cells. Biochim Biophys Acta. 1840, 730-738, 2014.

Kalyanaraman, B., Hardy, M., Podsiadly, R., Cheng, G., Zielonka, J.

Recent developments in detection of superoxide radical anion and hydrogen peroxide: opportunities, challenges, and implications in redox signaling. ABB, 617, 38-47, 2017

Ribou, AC

Synthetic sensors for reactive oxygen species detection and quantification: a critical review of current methods. ARS 25, 520-533, 2016.

Frijhoff J. et al.

Clinical relevance of biomarkers of oxidative stress. ARS 23, 1144-1170, 2015.

Dragsted LO

Biomarkers of exposure to vitamins A, C, and E and their relation to lipid and protein oxidation markers. Eur. J. Nutr 47, 3-18, 2008.

Schmidt HH et al.

Antioxidants in translational medicine. ARS 23, 1130-1143, 2015.

Author Response

Reviewer 3: At first glance I was very much looking forward to the promise announced in the abstract; reasons why vitamin E can have positive and adverse effects in the treatment of AD (by the way, treatment was not described, rather prevention of development).

We thank the reviewer for his/her deep revision of our manuscript. Thanks for this helpful and stimulating comment. We have added more details about the failure of vitamin E as treatment of AD:

“Another important point is bioavailability of vitamin E, which is complex and influenced by some important factors, such as the intake of competing nutrients, intestinal differences in absorption, age, gender, smoking, obesity and genetic polymorphisms. Sterols from plants (Richelle et al., 2004), eicosapentaenoic (Orneboe et al., 1990) and retinoic acids (Bieri et al., 1981) and even dietary fiber (Jenkins et al., 2001) are described as competing nutrients that decrease the absorption of vitamin E. Moreover, the rate of absorption into the bloodstream differs between 20% and 80% in humans, mainly due to the differences in food contents and the form of vitamin E. The transport of vitamin E in blood circulation is done by lipoproteins (Kolleck et al., 1999). In particular, the entrance to the central nervous system of -tocopherol in vivo is provided by HDL particles (Goti et al., 2000) and HDL levels vary greatly among people. In cells, vitamin E is bound to a-TTP, a transport protein highly expressed in AD brains and specially under oxidative stress conditions (Ulatowski et al., 2012). 

Age and gender also influence vitamin E bioavailability. After 60 years the levels of vitamin E increase in plasma but decrease after 80 years (Campbell et al., 1989). Women seem to have higher maximum plasma concentrations of -tocopherol than men (Leonard et al., 2005) maybe due to differences in HDL levels between genders.

 On the other hand, smokers have lower levels of a-tocopherol in serum than non-smokers (Al-Azemi et al., 2009; Shaah et al., 2015), although the variances could be attributed to the differences in dietary patterns (Galan et al., 2005). And obesity has an inverse relationship with serum α-tocopherol levels. In particular, the waist-to-hip ratio and waist circumference, were associated with α-tocopherol serum concentrations independently of gender (Wallström et al., 2001;  Ohrvall  et al., 1993; Ohrvall  et al., 1993).

Lastly, genetic polymorphisms in genes encoding for proteins involved in vitamin E metabolism could also explain interindividual differences in bioavailability (Döring et al., 2004; Borel et al., 2007, 2015; Major et al., 2011). More than 50 different SNPs has been related to vitamin E homeostasis. “

Likewise, we had in our original version a paragraph explaining both the prevention and also the main clinical trials performed on vitamin E as treatment. To clarify this we have added a subtitle: 3.4. Vitamin E and prevention of cognitive decline and in section  4 we have added to the title: 4. Is Vitamin E effective as treatment for AD? An approach to main trials.

However, reading the following was disappointing. The focus was laid only on the involvement of oxidative stress (free radicals) in the development of AD and the antioxidative property of vitamin E. (Free radical scavenging by vitamin E was never proved in vivo!) Related references were mainly derived from the last millennium. Thus, novelties in this direction were scarce. Methods used about 20 years ago to find oxidative stress as inducer or mediator of diseases were not specific and are no longer up-to-date (see references below).

We clarified this aspect in the “5. Why does vitamin E fail to treat AD?” section:

As we have discussed above, plasma levels of vitamin E are low in the patients, so we have evidences about the “baseline nutritional status”. But it is true that the measurement of baseline vitamin E levels is often not check in the main trials before beginning.

On the other hand, the extrapolation of results from animals to humans is not trivial.  The correct methods to measure oxidative stress are still under debate and sometimes are not specific or appropriated (Kalyanaraman et al., 2014; Zielonka et al., 2017; Ribou AC, 2016). Many studies of oxidative stress in AD are old and methods could not be up-to-date. To follow the antioxidant response by measuring an oxidative stress biomarker is very recommendable. However, to choose a good oxidative stress biomarker could not be easy. The specificity of some used biomarkers of oxidative stress, as in the case of oxidized low-density lipoprotein antibodies, could be questionable. In this regard, it has been recommended the analysis of oxidative stress in specific proteins involved in the disease. These markers could better represent a specific pathological pathway and a way for therapeutic monitoring and a prediction (Frijhoff et al., 2015) of results.

The focus on redox events in disease development is now concentrated on the redox-regulated modification of cellular signaling cascades involved in disease development or prevention. Modification of signaling pathways might well be a mechanism for the change in the other events hypothesized to start the AD development. According to the hypotheses provided in the introduction, the reader would expect a discussion in this direction.

We have now added to the result section that the detected Aβ appears mainly in cerebrovascular deposits, and discussed the role of Aβ peptides accumulating in the cerebrovasculature.

 Nevertheless, presenting the summary of available studies would help to further dig into the data and find a possible reason for the inconsistency of vitamin E effects. For this, the form of vitamin E, the doses, the time of intervention and the analytical methods should be included in all tables.

We have added the required information to the tables.

 In sum, in the present form, the manuscript cannot be recommended for publication. However, it might become more interesting if the state of the art is better presented.

References

·         Kalyanaraman, B., Darley-Usmar, V. Davies, K.J., Dennery, P.A., Forman, H.J., Grisham, M.B., Mann, G.E., Moore, K. Roberts II, L.J.,  Ischiropoulos, H.Measuring reactive oxygen and nitrogen species with fluorescence probes:  challenges and limitations. Free Radic. Biol. Med. 52:1-6, 2012. 

·         Winterbourn CC. The challenges of using fluorescent probes to detect and quantify specific oxygen species in living cells. Biochim Biophys Acta. 1840, 730-738, 2014. 

·         Kalyanaraman, B., Hardy, M., Podsiadly, R., Cheng, G., Zielonka, J.Recent developments in detection of superoxide radical anion and hydrogen peroxide: opportunities, challenges, and implications in redox signaling. ABB, 617, 38-47, 2017

·         Ribou, AC.Synthetic sensors for reactive oxygen species detection and quantification: a critical review of current methods. ARS 25, 520-533, 2016.

·         Frijhoff J. et al.Clinical relevance of biomarkers of oxidative stress. ARS 23, 1144-1170, 2015.

·         Dragsted LO. Biomarkers of exposure to vitamins A, C, and E and their relation to lipid and protein oxidation markers. Eur. J. Nutr 47, 3-18, 2008.

·         Schmidt HH et al. Antioxidants in translational medicine. ARS 23, 1130-1143, 2015.

We thank the reviewer for his/her comments and we hope that the manuscript is now better presented.

Round 2

Reviewer 3 Report

no further comments